# Stability Analysis of Continuous Stochastic Linear Model

**Jun Du** , **Bin Jia *** and **Shiteng Zheng**

Key Laboratory of Transport Industry of Big Data Application Technologies for Comprehensive Transport,
Ministry of Transport, Beijing Jiaotong University, Beijing 100044, China; 16114234@bjtu.edu.cn (J.D.);
18114018@bjtu.edu.cn (S.Z.)
* Correspondence: bjia@bjtu.edu.cn; Tel.: +86-138-1099-8385

**Abstract:** Many scholars have conducted research on the traffic oscillations and reproduced the growth pattern by establishing stochastic models and simulations. However, the growth pattern of oscillations caused by uncertainty have not been thoroughly studied. Recently, a frequency domain stability analysis method was proposed to analyze the discrete stochastic model. This paper extends this analysis to a continuous situation based on frequency domain tools (e.g., Laplace transform) by introducing a continuous bandlimited white noise. The analytical expression for the evolution of speed standard deviation has been derived. Our study of a homogeneous case reveals an interesting phenomenon: when $|G(\omega)|_\infty < 1$, the speed variance will converge to a constant value, which only depends on the self-disturbance of vehicles. The simulation results verified that the continuous models and corresponding discrete model tend to be consistent when the discrete time step tends to the infinitesimal. Overall, this paper makes up for the deficiency of previous studies on continuous oscillations in car-following theory and can potentially be used to develop new control strategies to help dampen traffic oscillations.

**Keywords:** car-following; traffic oscillations; stochastic analysis

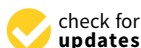

## 1. Introduction

In the last century, the research on traffic flow mainly focused on the classical models. The early exploration in this field can be traced back to 1953. Pipes [1] used a differential dynamics equation to describe car-following behavior and assumed that there is a positive correlation between the speed difference and the following vehicle acceleration, in other words, when the speed of the preceding vehicle is greater/lower than that of the following vehicle, the following vehicle will accelerate/decelerate. Newell [2] took the distance between the preceding and following vehicle into account and proposed that the speed of the following vehicle is positively correlated with the distance. Furthermore, Bando et al. [3] proposed that there exists an 'optimal velocity' determined by that distance. He also proposed that the acceleration of the following vehicle should be related to the 'optimal velocity' and the current velocity. Based on this optimal velocity model, Hilbing and Tilch [4] developed a generalized force model by considering the negative velocity difference to avoid unrealistic deceleration. Jiang et al. [5] considered both negative and positive velocity difference and developed a full velocity model. A nonlinear analysis method was proposed by Xue et al. [6] with a full velocity model by which a reasonable kinematics wave speed can be obtained. Zhao et al. [7] accounted for acceleration and proposed a full velocity and acceleration difference model (FVADM). Nagatani [8–10] and Sawada [11] extended the car-following model with a next-nearest neighbor interaction.

Although these classic models have clear and elegant stability properties, the growth pattern of oscillation is inconsistent with the field experiment, in which the speed standard deviation of each vehicle developed concavely. Jiang et al. established 2D models by changing the deterministic parameters to stochastic and successfully simulated the concave pattern in 2014 [12]. Lang et al. [13] improved the inertia model and proposed that the

amplitudes of car accelerations become larger along the car platoon; Xu et al. [14] presented an analysis of a Newell-type stochastic car-following model based on the stochastic desired acceleration processes. After which, a number of stochastic car-following models were developed [15–18].

From another point of view, bottlenecks and lane-changing have been believed to be the main causes of traffic oscillations [19–22] until Sugiyama's experiment [23], which shows traffic oscillations occurred in the absence of lane changing/bottlenecks. However, researchers are still in debate about traffic oscillations being a result of the string instabilities in the mathematical models [24–26] or heterogeneous driving behavior [27–29]. This paper put forward another view that the traffic oscillations are caused not by the model or heterogeneous behavior, but by the disturbance of the drivers/vehicles itself.

In 2020, Wang et al. [30] proposed the frequency-domain stability analysis method for linear stochastic car-following models, extending the traditional frequency domain analysis tool of the deterministic model to stochastic models. This method is able to quantify speed variations of a stream of vehicles following one another according to certain stochastic linear car-following behaviors. However, this method is only applicable in a time-discrete condition, which is inconsistent with the fact that physical time should be continuous. Motivated by the fact, this paper extends the abovementioned method to the first-order model under time-continuous circumstance to verify whether this method is still feasible. We also discussed the feasibility for extending this model to second-order models. Our study of the homogeneous case reveals that when traffic is stable, the speed variance converges to a constant value, which only depends on the self-disturbance of drivers/vehicles. The simulation results verified that the continuous models and corresponding discrete model tend to be consistent when discrete time step tends to the infinitesimal. The continuous models and corresponding discrete model tend to be consistent in our simulations. Overall, this paper fills the gap in the research of continuous noise in traffic-flow theory and could potentially be used to develop new control strategies to help in dampening traffic oscillations.

The organization of this paper is as follows. In Section 2, the continuous stochastic linear First-order model is described. In Section 3, the stability analysis method is performed and reveals the relationship between the discrete and continuous model. In Section 4, numerical examples are provided to illustrate the application of the proposed method to continuous stochastic linear first-order models. Section 5 concludes this paper and discusses future research directions.

## 2. Model

Assuming that a vehicle platoon with an index of n = 1,2,3 . . . , N moves on a single lane without any overtaking as shown in Figure 1, we use $\mathcal{N}$ to denote the index set {1,2,3 . . . , N} for convenience.

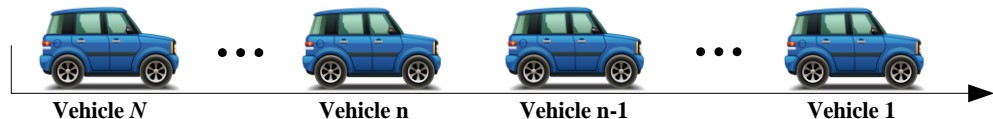

**Figure 1.** Illustration of the car-following system with N vehicles in the system.

We introduce the definition of string stability as follows: the string of a vehicle is stable if, for any set of bounded initial disturbances to all the vehicles, the position fluctuations of all the vehicles remain bounded [31]. For the convenience of the readers, the key notation is summarized in Table 1.

**Table 1.** Notation List.

| Notation | Definition | Notation | Definition |
|---|---|---|---|
| N | total vehicle number | $r_1(\xi)$ | the autocorrelation function of $v_1(t)$ |
| $\mathcal{N}$ | vehicle index set | $S_1(\omega)$ | power density |
| $v_n^0(t)$ | the speed of the $n$th vehicle at time $t$ | $V_n(s)$ | Laplace transformation of $v_n(t)$ |
| $p_n^0(t)$ | the position of the $n$th vehicle at time $t$ | $G_n(s)$ | transfer function between the $n-1$ and $n$th vehicle |
| $v_n(t)$ | the speed of the $n$th vehicle at time $t$ after linearization | $\overline{\Delta}_n$ | speed variance of the $n$th vehicle under DFM |
| $p_n(t)$ | the position of the $n$th vehicle at time $t$ after linearization | $\Delta_n$ | speed variance of the $n$th vehicle under SFM |
| $\alpha_n$ | Constant coefficient | $\hat{\Delta}_n$ | simulate speed variance under DFM |
| $\tau$ | reaction time | $\widetilde{\Delta}_n$ | simulate speed variance under SFM |
| $\varepsilon_n(t)$ | continuous bandlimited white noise | $\acute{\Delta}_n$ | noise term (continuous condition) |
| $\sigma_n^2$ | variance of white noise $\varepsilon_n(t)$ | $\grave{\Delta}_n$ | noise term (Discrete condition) |

We use $v_n^0(t)$, $p_n^0(t)$ to denote the speed and position of the $n$th vehicle at time $t$. Assuming that at the initial time $t = 0$, all vehicles are in a stationary condition, which implies the same starting speed $\overline{v}$ and corresponding spacing $\overline{p}_n$, i.e.:

$$v_n^0(0) = \overline{v}, \ p_{n-1}^0(0) - p_n^0(0) = \overline{p}_n, \ \forall n \in \mathcal{N} \tag{1}$$

Since this article focuses on the study of oscillation, we convert the variables as follows according to the predecessor's method [30]:

$$v_n(t) := v_n^0(t) - \overline{v}, \ p_n(t) := p_n^0(t) - p_n^0(0) - \overline{v}t, \ \forall n \in \mathcal{N} \tag{2}$$

Assuming all vehicles follow the stochastic first-order model (SFM):

$$\text{SFM}: v_n(t + \tau) = \alpha_n \left(p_{n-1}(t) - p_n(t)\right) + \varepsilon_n(t), \quad \forall n \in \mathcal{N} \backslash 1 \tag{3}$$

where $\tau$ is the reaction time and the definition of $\varepsilon_n(t)$ is as follows: $\varepsilon_n(t)$ is a continuous bandlimited white noise with a variance of $\sigma_n^2$ defined on $(0 \sim T), (T \to \infty)$ with equal spectral density $E_n$ at different angular frequencies $(0 \sim \mathbb{F}), (\mathbb{F} \to \infty)$, which means $\varepsilon_n(t)$ satisfies the following equations:

$$\lim_{T \to \infty} \frac{1}{T} \int_0^T |\varepsilon_n(t)|^2 dt = \sigma_n^2 \tag{4}$$

$$|E_n(\omega)| = \begin{cases} E_n, & if \ 0 \leq \omega < F \\ 0, & otherwise \end{cases} \tag{5}$$

where $E_n(\omega)$ is the Fourier transform of $\varepsilon_n(t)$, i.e.: $E_n(\omega) := \lim_{T \to \infty} \int_0^T \varepsilon_n(t)e^{-j\omega t}dt$. It is worth noting that the reason we do not use a continuous unlimited white noise is that an unlimited white noise has infinite variance, which is not realistic. To analyse the SFM, we define the deterministic first-order model (DFM) as follows:

$$\text{DFM}: v_n(t + \tau) = \alpha_n \left(p_{n-1}(t) - p_n(t)\right), \quad \forall n \in \mathcal{N} \backslash 1 \tag{6}$$

## 3. Stability Analysis

### 3.1. Deterministic Stability Analysis

In this subsection, the task is to obtain the speed variance of the $n$th vehicle under DFM. The Laplace transformation of $v_n(t)$ can be written as follows:

$$V_n(s) \int_0^\infty v_n(t)e^{-st}dt \tag{7}$$

Applying the Laplace transformation to both sides of Equation (6), we can obtain the Laplace form of DFM as follows:

$$(se^{\tau s} + \alpha_n)V_n(s) = \alpha_n V_{n-1}(s), \quad \forall n \in \mathcal{N} \backslash 1 \tag{8}$$

Thus, the relation between the $(i-1)$th vehicle velocity disturbance and the $i$th vehicle velocity disturbance is described by:

$$V_n(s) = G_n(s)V_{n-1}(s), \quad \forall n \in \mathcal{N} \backslash 1 \tag{9}$$

where the transfer function $G_n(s)$ of DFM is:

$$G_n(s) = \frac{\alpha_n}{se^{\tau s} + \alpha_n} \tag{10}$$

According to Parseval's theorem, we could obtain

$$\lim_{T \to \infty} \int_0^T |v_n(t)|^2 dt = \frac{1}{2\pi} \int_{-\infty}^{\infty} |V_n(j\omega)|^2 d\omega \tag{11}$$

where $j$ is the imaginary unite and $s = j\omega$. We use $V_n(\omega)$ to denote $V_n(j\omega)$ in the following. Dividing both sides of Equation (11) by $T(T \to \infty)$, we can transform Parseval's theorem to the following form (which is also the definition of average power $P_n$):

$$\lim_{T \to \infty} \int_0^T |v_n(t)|^2 dt = \frac{1}{2\pi} \int_{-\infty}^{\infty} |V_n(j\omega)|^2 d\omega \tag{12}$$

$$P_n = \lim_{T \to \infty} \frac{1}{T} \int_0^T |v_n(t)|^2 dt = \frac{1}{2\pi} \int_{-\infty}^{\infty} \lim_{T \to \infty} \frac{1}{T} |V_n(\omega)|^2 d\omega \tag{13}$$

Noting that in the middle part of Equation (13) is the speed variance of the $n$th vehicle under DFM, and $\lim_{T \to \infty} \frac{1}{T} |V_n(\omega)|^2$ is the power density of $v_n(t)$, i.e.:

$$\overline{\Delta}_n \lim_{T \to \infty} \frac{1}{T} \int_0^T |v_n(t)|^2 dt \tag{14}$$

$$S_n(\omega) := \lim_{T \to \infty} \frac{1}{T} |V_n(\omega)|^2 \tag{15}$$

we can denote Equation (13) as:

$$\overline{\Delta}_n = \frac{1}{2\pi} \int_{-\infty}^{\infty} S_n(\omega) d\omega \tag{16}$$

and also, we can obtain:

$$S_n(\omega) = \prod_{m=2}^{n} |G_m(\omega)|^2 S_1(\omega) \tag{17}$$

where $S_1(\omega)$ can be obtained by the Wiener–Khinchin theorem [32]: the autocorrelation function has a spectral decomposition provided by the power spectrum of that process, i.e.:

$$S_1(\omega) = \int_{-\infty}^{\infty} r_1(\xi) e^{-j\omega\xi} d\xi \tag{18}$$

where $r_1(\xi)$ is the autocorrelation function of $v_1(t)$, i.e.,:

$$r_1(\xi) E[v_1(t) \cdot v_1(t + \xi)] \tag{19}$$

To achieve $r_1(\xi)$, we set three leading vehicle cases as follows:

Case 1: $v_1(t) = A_0 \sin(\omega_0 t)$:

$$r_1(\xi) = E[A_0 \sin(\omega_0 t) \cdot A_0 \sin(\omega_0(t+\xi))] = \frac{A_0^2}{2} \cos(\omega_0 \xi) \tag{20}$$

Then, we can easily obtain $S_1(\omega)$ and $\overline{\Delta}_n$ through Equations (15), (16) and (18):

$$S_1(\omega) = \frac{A_0^2 \pi}{2}(\delta(\omega - \omega_0) + \delta(\omega + \omega_0)) \tag{21}$$

$$\overline{\Delta}_n = \frac{A_0^2}{2} \prod_{m=2}^{n} |G_m(\omega_0)|^2 \tag{22}$$

Case 2: $v_1(t)$ is a finite summation of the trigonometric functions:

$$v_1(t) = \sum_{k=1}^{K} a_k \sin(\omega_k t) \quad (\omega_k \neq \omega_l \ if \ k \neq l) \tag{23}$$

The autocorrelation function can be derived as follows, according to the orthogonality of trigonometric functions:

$$r_1(\xi) = \sum_{k=1}^{K} E[a_k \sin(\omega_k t) \cdot a_k \sin(\omega_k(t+\xi))] = \sum_{k=1}^{K} \frac{a_k^2}{2} \cos(\omega_k \xi) \tag{24}$$

Then, $S_1(\omega)$ can be derived using Equation (18), and $\overline{\Delta}_n$ can be derived using Equations (16) and (17):

$$S_1(\omega) = \sum_{k=1}^{K} \frac{a_k^2 \pi}{2}(\delta(\omega - \omega_k) + \delta(\omega + \omega_k)) \tag{25}$$

$$\overline{\Delta}_n = \sum_{k=1}^{K} \frac{a_k^2}{2} \prod_{m=2}^{n} |G_m(\omega_k)|^2 \tag{26}$$

From the above equation, we can derive the following proposition:

**Proposition 1:** *If the leading vehicle moves as $v_1[i] = \sum_{k=1}^{K} f_k[i]$, and $f_k[i]$ are orthogonal, then*

$$\overline{\Delta}_n = \sum_{k=1}^{K} \overline{\Delta}_{nk} \tag{27}$$

*where $\overline{\Delta}_{nk}$ is the speed variance of the nth vehicle with the leading vehicle move as $f_k[i]$ under DFM.*

Case 3: $v_1(t)$ is a bandlimited white noise as defined in Section 2, with a variance of $\sigma_1^2$:

By using definition of the Fourier transform of $\varepsilon_n(t)$, i.e., $E_n(\omega) := \lim_{T \to \infty} \int_0^T \varepsilon_n(t)e^{-j\omega t}dt$, according to Parseval's theorem, we can obtain:

$$\lim_{T \to \infty} \int_0^T |\varepsilon_n(t)|^2 dt = \frac{1}{2\pi} \int_{-\infty}^{\infty} |E_n(\omega)|^2 d\omega \tag{28}$$

Speed variance can be derived by dividing both sides by $T$:

$$\sigma_n^2 = \lim_{T \to \infty} \frac{1}{T} \int_0^T |\varepsilon_n(t)|^2 dt = \lim_{T \to \infty} \frac{1}{2\pi T} \int_{-\infty}^{\infty} |E_n(\omega)|^2 d\omega \tag{29}$$

According to Equation (5), we can derive the following equation:

$$\lim_{T\to\infty}\frac{1}{2\pi T}\int_{-\infty}^{\infty}|E_n(\omega)|^2 d\omega = \lim_{T\to\infty}\lim_{\mathbb{F}\to\infty}\frac{1}{2\pi T}\int_0^{\mathbb{F}}|E_n|^2 d\omega \tag{30}$$

Therefore:

$$\sigma_n^2 = \lim_{T\to\infty}\lim_{\mathbb{F}\to\infty}\frac{1}{2\pi T}\int_0^{\mathbb{F}}|E_n|^2 d\omega \Rightarrow |E_1|^2 = \lim_{T\to\infty}\lim_{\mathbb{F}\to\infty}\frac{2\pi T\sigma_1^2}{\mathbb{F}} \tag{31}$$

$$S_1(\omega) = \lim_{T\to\infty}\frac{1}{T}|E_1(\omega)|^2 = \begin{cases} \lim_{\mathbb{F}\to\infty}\frac{\sigma_n^2 2\pi}{\mathbb{F}}, & if\ \ 0 \le \omega < F \\ 0, & otherwise \end{cases} \tag{32}$$

Then, we can easily and $\overline{\Delta}_n$ through Equations (15) and (16):

$$\overline{\Delta}_n = \frac{1}{2\pi}\int_{-\infty}^{\infty}\prod_{m=2}^{n}|G_m(\omega)|^2 S_1(\omega)d\omega = \lim_{\mathbb{F}\to\infty}\frac{\sigma_n^2}{\mathbb{F}}\int_0^{\mathbb{F}}\prod_{m=2}^{n}|G_m(\omega)|^2 d\omega \tag{33}$$

It is obvious that $|G_n(\omega)|^2 \to 0$ as $\omega \to \infty$, thus $\overline{\Delta}_n = \lim_{\mathbb{F}\to\infty}\frac{\sigma_n^2}{\mathbb{F}}\int_0^{\mathbb{F}}\prod_{m=2}^{n}|G_m(\omega)|^2 d\omega = 0$ when $n \ge 2$.

### 3.2. Stochastic Stability Analysis

By applying Fourier transform to the SFM Equation (3), we can obtain the frequency-domain equation as:

$$V_n(\omega) = V_{n-1}(\omega)G_n(\omega) + H(\omega)E_n(\omega) \tag{34}$$

where $H(\omega) = \frac{j\omega}{j\omega e^{\tau j\omega} + \alpha_n}$, with recursion to Equation (28), we obtain:

$$V_n(\omega) = \prod_{m=2}^{n}G_m(\omega)V_1(\omega) + \sum_{m=2}^{n}\prod_{m'=m+1}^{n}G_{m'}(\omega)\cdot H(\omega)E_n(\omega) \tag{35}$$

Therefore, the power density $S_n(\omega)$ of $v_n[i]$ can be formulated as:

$$S_n(\omega) = \prod_{m=2}^{n}|G_m(\omega)|^2 S_1(\omega) + \lim_{T\to\infty}\frac{1}{T}\sum_{m=2}^{n}\prod_{m'=m+1}^{n}|G_{m'}(\omega)|^2\cdot|E_m(\omega)H(\omega)|^2 \tag{36}$$

Because $\Delta_n = \frac{1}{2\pi}\int_{-\infty}^{\infty}S_n(\omega)d\omega$, the speed variance can be expressed as follows:

$$\Delta_n = \overline{\Delta}_n + \lim_{\mathbb{F}\to\infty}\frac{\sigma_n^2}{\mathbb{F}}\int_0^{\mathbb{F}}\sum_{m=2}^{n}\prod_{m'=m+1}^{n}|G_{m'}(\omega)|^2\cdot|H(\omega)|^2 d\omega \tag{37}$$

where $\overline{\Delta}_n$ can be found in Equations (22), (26) and (33) for all three cases.

### 3.3. Discussion on Second Order Model

In this section, we apply the above method to second order models to analyze feasibility. To achieve the above purpose, we use the stochastic linear optimal velocity model [3] as an example:

$$a_n[i + d_n] = \beta_n(\alpha_n\ (p_{n-1}[i] - p_n[i]) - v_n[i]) + \varepsilon_n(t), \forall n \in \mathcal{N}\backslash 1 \tag{38}$$

Thus, we can conduct the transfer function and the speed variance of the $n$th vehicle under the Homogeneous case as follows:

$$G(s) = \frac{\alpha\beta}{e^{\tau s}s^2 + \beta s + \alpha\beta} \tag{39}$$

$$\Delta_n = \overline{\Delta}_n + \acute{\Delta}_n \tag{40}$$

where $\acute{\Delta}_n = \lim_{\mathbb{F}\to\infty} \frac{\sigma^2}{\mathbb{F}} \int_0^{\mathbb{F}} \frac{\left(1-|G(\omega)|^{2(n-1)}\right)\cdot|H(\omega)|^2}{\left(1-|G(\omega)|^2\right)} d\omega$ and $H(\omega) = \frac{j\omega}{-e^{\tau s}\omega^2 + \beta j\omega + \alpha\beta}$, it is obvious that $|H(\omega)| \to 0$ and $(1-|G(\omega)|^2) \to 1$ as $\omega \to \infty$, thus $\acute{\Delta}_n = 0$ for $n \in \mathcal{N}$, which means that the stochastic term has no effect on the speed variance under the stochastic linear optimal velocity model. The reason for the different effects is that the transfer function has a second order term for the denominator of the second order model, which leads to $|H(\omega)| \to 0$ as $\omega \to \infty$, but for the first-order model, $|H(\omega)| \to 1$ as $\omega \to \infty$.

### 3.4. Homogeneous Case Analysis

We assume that all the vehicles have the same transfer function and white noise factor, which means:

$$G_n(\omega) = G(\omega), \ \forall n \in \mathcal{N}\backslash 1 \tag{41}$$

$$\sigma_n^2 = \sigma^2 \ E_n = E, \ \forall n \in \mathcal{N}\backslash 1 \tag{42}$$

By plugging Equations (41) and (42) to Equations (22), (26), (33) and (37), the speed variance for the homogeneous scene under DFM and SFM could be expressed as follows:

$$\overline{\Delta}_n = \frac{A_0^2}{2}|G(\omega_0)|^{2(n-1)} \text{(Case 1 for DFM)} \tag{43}$$

$$\overline{\Delta}_n = \sum_{k=1}^{K} \frac{a_k^2}{2}|G(\omega_k)|^{2(n-1)} \text{(Case 2 for DFM)} \tag{44}$$

$$\overline{\Delta}_n = \begin{cases} 0 & if \ n \geq 2 \\ \sigma_1^2 & if \ n = 1 \end{cases} \text{(Case 3 for DFM)} \tag{45}$$

$$\Delta_n = \overline{\Delta}_n + \lim_{\mathbb{F}\to\infty} \frac{\sigma_n^2}{\mathbb{F}} \int_0^{\mathbb{F}} \sum_{m=2}^{n} |G(\omega)|^{2(n-m)}\cdot|H(\omega)|^2 d\omega \text{(Cases 1–3 for SFM)} \tag{46}$$

Noticing $|G(\omega)|^{2(n-m)}\cdot|H(\omega)|^2$ is a geometric series, we obtain:

$$\Delta_n = \overline{\Delta}_n + \lim_{\mathbb{F}\to\infty} \frac{\sigma^2}{\mathbb{F}} \int_0^{\mathbb{F}} \frac{\left(1 - |G(\omega)|^{2(n-1)}\right)\cdot|H(\omega)|^2}{\left(1 - |G(\omega)|^2\right)} d\omega \tag{47}$$

**Proposition 2:** *From the above equation, it is obvious that when $|G(\omega)|_\infty > 1$, $\Delta_n \to \infty$ as $n \to \infty$; when $|G(\omega)|_\infty < 1$, $\Delta_n \to \lim_{\mathbb{F}\to\infty} \frac{\sigma^2}{\mathbb{F}} \int_0^{\mathbb{F}} \left|\frac{1}{e^{\tau\omega j}}\right| d\omega = \sigma^2$ as $n \to \infty$, which reveals that the traffic oscillations is caused only by the disturbance of the drivers/vehicles itself.*

### 3.5. Relationship between Discrete and Continuous Model

In this subsection, we analyze the difference between the discrete and continuous first-order model under homogeneous case, and list the discrete and continuous first-order model as follows:

First-order discrete model:

$$v_n[i + d_n] = \alpha_n \ (p_{n-1}[i] - p_n[i]) + \varepsilon_n[i], \forall n \in \mathcal{N}\backslash 1 \tag{48}$$

First-order continuous model:

$$v_n(t + \tau) = \alpha_n \ (p_{n-1}(\tau) - p_n(\tau)) + \varepsilon_n(t), \forall n \in \mathcal{N}\backslash 1 \tag{49}$$

Predecessors already conducted the standard deviation for the discrete model in previous work [30], the expression of standard deviation in the discrete model has a similar

expression under a continuous model for Case 1, but the transfer function has different expressions under the discrete/continuous model:

$$\overline{G}(\omega) = \frac{\alpha\delta}{e^{j\omega d} - e^{j\omega(d-1)} + \alpha\delta}, \omega\epsilon(0, 2\pi] \text{(Discrete model)} \tag{50}$$

$$G(\omega) = \frac{\alpha}{j\omega e^{\tau j\omega} + \alpha}, \omega\epsilon(0, \infty) \text{(Continuous model)} \tag{51}$$

where $\delta$ is the discretization time interval and $d\delta > 0$ is vehicle $n$'s reaction time $\tau$. Figure 2 shows the relationship between $\omega$ and $G(\omega)$ under the Continuous model, as well as $\omega/\delta$ and $\overline{G}(\omega)$ under the Discrete model. The parameters are as follows: $\alpha = 0.5\, \tau = 1.2\, d = \tau/\delta$. One can see that $\overline{G}(\omega) \to G(\omega)$ as $\delta \to 0$.

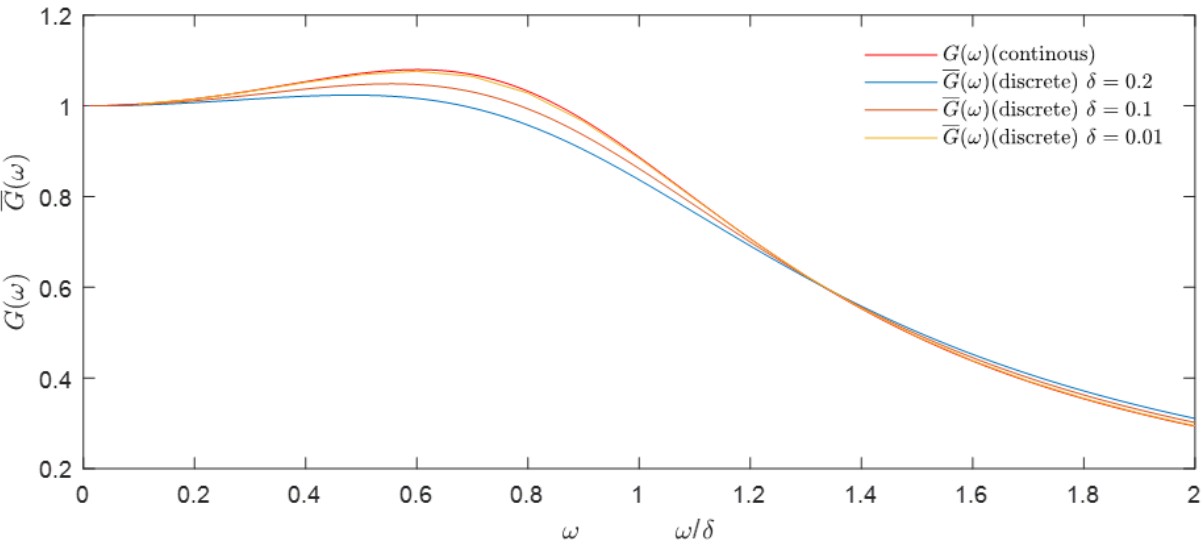

**Figure 2.** Magnitudes of $G(\omega)$ and $\overline{G}(\omega)$.

We also compare the noise term $\acute{\Delta}_n = \frac{\sigma^2}{\mathbb{F}} \int_0^{\mathbb{F}} \frac{\left(1-|G(\omega)|^{2(n-1)}\right) \cdot |H(\omega)|^2}{\left(1-|G(\omega)|^2\right)} d\omega$ (Continuous model) with $\grave{\Delta}_n = \frac{\sigma^2}{2\pi} \int_0^{2\pi} \frac{\left(1-|G(\omega)|^{2(n-1)}\right)}{|L_1(\omega)|^2 \left(1-|G(\omega)|^2\right)} d\omega$ (Discrete model) in Figure 3. Parameters are as follows: $\mathbb{F} = 10, 100, 1000; \sigma^2 = 1; \delta = 0.1, 0.01, 0.001; \alpha = 0.3\, \tau = 1.4$. One can see that $\acute{\Delta}_n \to \grave{\Delta}_n$ as $\mathbb{F} \to \infty$ and $\delta \to 0$.

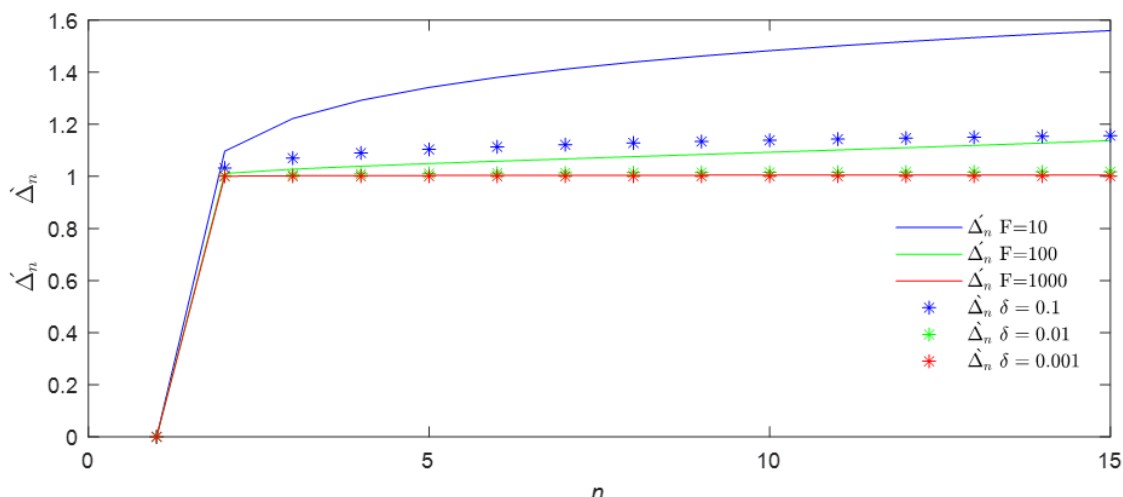

**Figure 3.** Noise term comparing between continuous and discrete model.

In this section, we simulatively show that when a discrete interval $\delta \to 0$, the transfer function of discrete model converges to that of the continuous model and furthermore, when frequency $\mathbb{F} \to \infty$, the noise term of discrete function and continuous function converge to each other. Thus, we can derive that $\Delta_{n,discrete} \to \Delta_n$ when $\mathbb{F} \to \infty$ and $\delta \to 0$. (Simulation shows in Appendix A)

### 4. Numerical Simulations

In this section, we let the leading vehicle move as $v_1(t) = A_1 \sin(\omega_1 t) + A_2 \sin(\omega_2 t)$, $(\omega_1 \neq \omega_2)$, then a speed variance of DFM and SFM can be driven through Equations (44) and (46):

$$\overline{\Delta}_n = \frac{A_1^2}{2} \left| G\left( \frac{\alpha}{j\omega_1 e^{\tau j\omega_1} + \alpha} \right) \right|^{2(n-1)} + \frac{A_2^2}{2} \left| G\left( \frac{\alpha}{j\omega_2 e^{\tau j\omega_2} + \alpha} \right) \right|^{2(n-1)} \tag{52}$$

$$\Delta_n = \overline{\Delta}_n + \frac{\sigma_n^2}{\mathbb{F}} \int_0^{\mathbb{F}} \sum_{m=2}^{n} \left| \frac{\alpha}{j\omega e^{\tau j\omega} + \alpha} \right|^{2(n-m)} \cdot \left| \frac{j\omega}{j\omega e^{\tau j\omega} + \alpha_n} \right|^2 d\omega \tag{53}$$

To verify Proposition 1, we decompose $v_1(t) = v_{1,1}(t) + v_{2,1}(t)$, where $v_{1,1}(t) = A_1 \sin(\omega_1 t)$ and $v_{2,1}(t) = A_2 \sin(\omega_2 t)$, and we denote the simulation result of speed variance with the leading speed $v_{1,1}(t)$ and $v_{2,1}(t)$ under DFM as $\hat{\Delta}_{1,n}$ and $\hat{\Delta}_{2,n}$. We use Equation (46) as the discrete model, $\delta = 0.001$ as a very tiny discrete time interval to simulate the traffic flow system and $\hat{\Delta}_n \sum_{i=1+I/5}^{I} (v_n[i])^2$ to denote the simulation speed variance under DFM to avoid the impact of start-up, where $I = 10^6$. Correspondingly, we run the simulation for $M = 1000$ iterations under SFM, and directly measure the speed variance of SFM using the following equation:

$$\widetilde{\Delta}_n \frac{\sum_{m=1}^{M} \sum_{i=1+I/5}^{I} (v_n[i])^2}{4MI/5} \tag{54}$$

The simulation settings are taken as follows: $\mathbb{F} = 3000$, $N = 30$, $\tau = 1$, $d = \tau/\delta$, $\alpha = 0.5$, $\sigma = 2$, $A_1 = 1.5$, $\omega_1 = 1$, $A_2 = 1$, $\omega_2 = 2$. As the results demonstrate in Figure 4, we can see $G(\omega_1) < 1$ and $G(\omega_2) < 1$ from Figure 4b, thus DFM with a leading speed of $v_1(t)$ is stable. We can see that $\sqrt{\hat{\Delta}_{1,n} + \hat{\Delta}_{2,n}}$, $\sqrt{\overline{\Delta}_n}$ and $\sqrt{\hat{\Delta}_n}$ overlapped very well with each other, which is consistent with Proposition 1. Additionally, $|G(\omega)|_\infty < 1$, thus $\sqrt{\Delta_n} \to 2$, is consistent with Proposition 2.

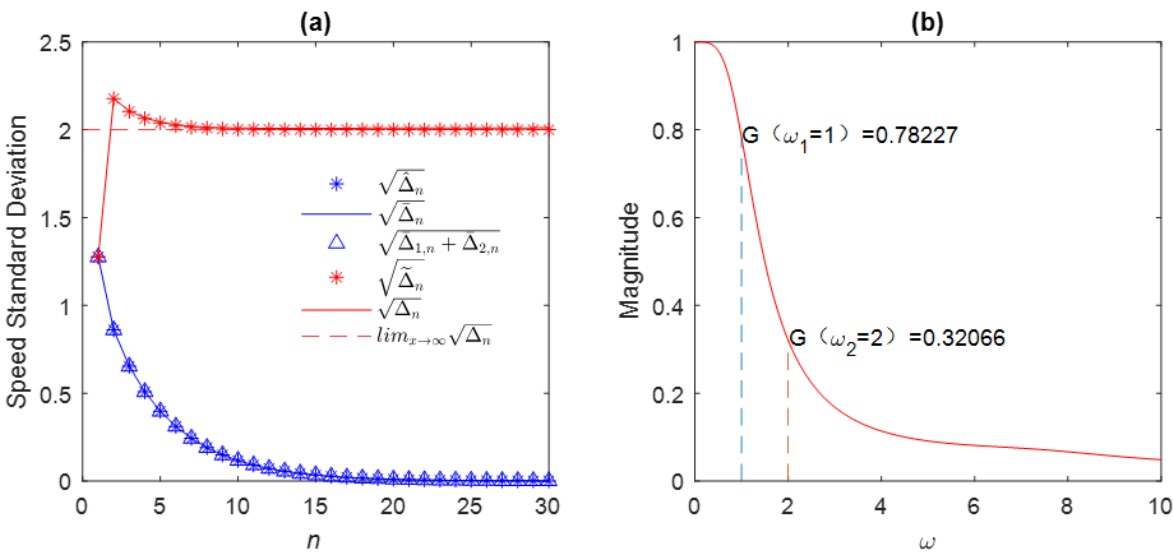

**Figure 4.** Simulation results with $\alpha = 0.5, \sigma = 2, \tau = 1$: (**a**) Time-Domain Speed Standard Deviation; (**b**) Magnitudes of $G(\omega)$, $G(\omega_1)$ and $G(\omega_2)$.

Then, we exam this case with $\tau = 1.5$. The simulation result is shown in Figure 5. It is obvious that $|G(\omega)|_{\infty} > 1$ from Figure 5b, thus $\Delta_n \to \infty$ as $n \to \infty$, which is consistent with Proposition 2.

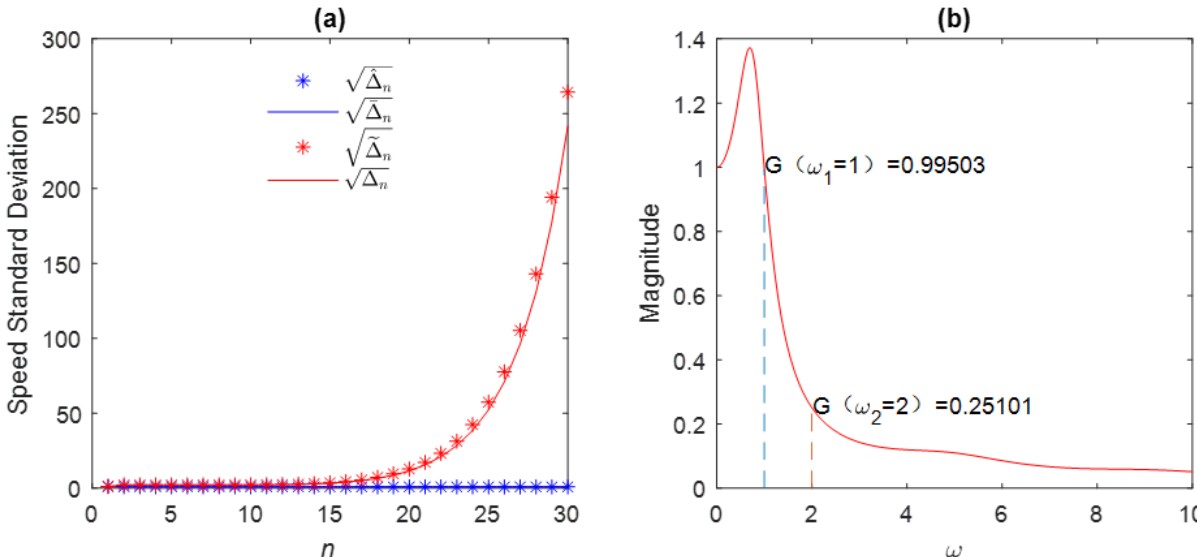

**Figure 5.** Simulation results with $\alpha = 0.5$, $\sigma = 2$, $\tau = 1.5$: (**a**) Time-Domain Speed Standard Deviation; (**b**) Magnitudes of $G(\omega)$, $G(\omega_1)$ and $G(\omega_2)$.

To better correlate with field data, the GPS data from our previous work were utilized [33]. The sampling frequency of the GPS data is 0.1s. We randomly selected the speed track of one vehicle as the speed data of the leading vehicle. The data length is 930.7s, which means $T = 9307$ sampling points, see Figure 6.

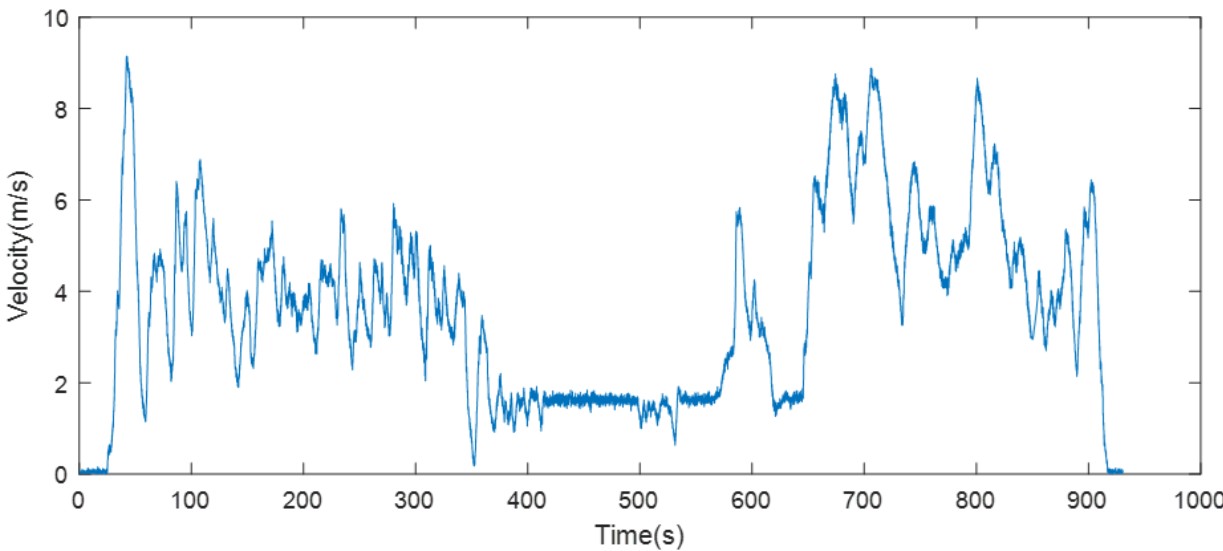

**Figure 6.** Leading vehicle speed data.

We decompose $v_1[i]$ as $v_1[i] = \sum_{k=1}^{T} a_k \sin\left[\frac{k}{T} 2\pi i\right] + b_k \cos\left[\frac{k}{T} 2\pi i\right]$, where $a_k = \frac{1}{T} \sum_{i=1}^{T} v[i]$ $\cos\left(\frac{k}{T} 2\pi i\right)$, $b_k = \frac{1}{T} \sum_{i=1}^{T} v[i] \sin\left(\frac{k}{T} 2\pi i\right)$ and $T = 9307$. Then, we recompose $v_1(t) = \sum_{k=1}^{T} a_k \sin\left[\frac{k}{T} 2\pi t\right] + b_k \cos\left[\frac{k}{T} 2\pi t\right]$ as the leading vehicle. To avoid the impact of start-ups, we used two cycles of the leading vehicle in the simulation. The simulation settings are as follows: $\mathbb{F} = 3000$, $\delta = 0.001$, $N = 30$, $\tau = 1$, $\alpha = 0.5$, $\sigma = 2.5$. The simulation result is

shown in Figure 7, where the simulation results and the theoretical results overlap with each other and $\sqrt{\Delta_n} \to 2.5$, which is consistent with Proposition 2.

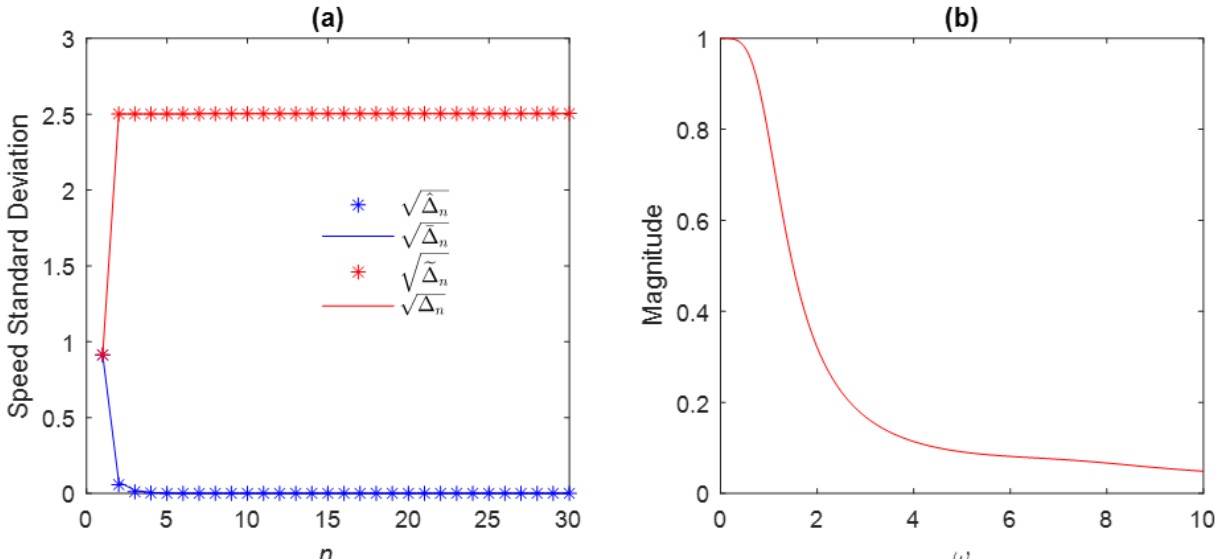

**Figure 7.** Simulation results with $\alpha = 0.5, \sigma = 2.5, \tau = 1$: (**a**) Time-Domain Speed Standard Deviation; (**b**) Magnitudes of $G(\omega)$.

## 5. Conclusions

In this paper, we extend the previous work [30] to the Fist-order continuous model, which allows us to analytically quantify the speed variance without discretize the model, the main contribution of this paper is as follows:

-   This paper presents an analytical expression for the evolution of speed standard deviation under a continuous first-order model.
-   We simulatively show that when the discrete interval δ→0, the transfer function of the discrete model and continuous model converge to each other; when frequency F→∞, the noise term of discrete function and continuous function converge to each other.
-   This paper put forward another view that traffic oscillations are caused not by the model or heterogeneous behavior, but by the disturbance of the drivers/vehicles. From the traffic control aspect, this paper reveals that the key point of dampening the highway traffic oscillations is to reduce the noise caused by driver/vehicles. (Specific measures can be to improve driving concentration, pavement repair, and reduce roadside distractions, etc.)

This study proposed an analytical method for the first-order continuous stochastic model, but there is still work that needs to be done. Firstly, white noise may not be the best choice of the stochastic term; therefore, we need to further analyze the form of noise. Secondly, we need to study the relationship between acceleration noise and velocity noise to extend the method to models such as the optimal velocity model or full velocity difference model. Finally, this method might be potentially extended to nonlinear stochastic models using the describing function method proposed in Li [34].

**Author Contributions:** Conceptualization, J.D. and B.J.; methodology, J.D.; validation, J.D. and S.Z.; formal analysis, J.D.; investigation, J.D.; data curation, J.D.; writing—original draft preparation, J.D.; writing—review and editing, B.J.; visualization, S.Z.; supervision, B.J.; project administration, B.J.; funding acquisition, B.J. All authors have read and agreed to the published version of the manuscript.

**Funding:** This research was funded by the National Natural Science Foundation of China, grant number 71971015 and 71621001 and the Research Foundation of the state key Laboratory of Rail Traffic Control and Safety, grant number RCS2020ZI001.

**Institutional Review Board Statement:** Not applicable.

**Informed Consent Statement:** Not applicable.

**Data Availability Statement:** The data are contained within the article.

**Conflicts of Interest:** The authors declare no conflict of interest.

**Appendix A**

Proposition:

$$\Delta_{n,discrete} \to \Delta_n \ \text{ when } F \to \infty \ \text{ and } \delta \to 0.$$

Simulation:

We set the leading vehicle movement as $v_1(t) = A_0 \sin(\omega_0 t)$ (for the continuous model) and $v_1[i] = A_0 \sin(\omega_0 i \delta)$ (for the discrete model). According to previous work, the speed variance of the nth vehicle under discrete model can be derived as:

$$\Delta_{n,discrete} = \frac{A_0^2}{2} \left| \frac{\alpha \delta}{e^{j\omega_0 \delta d} + e^{j\omega_0 \delta(d-1)} + \alpha \delta} \right|^{2(n-1)}$$
$$+ \frac{1}{2\pi} \int_0^{2\pi} \sum_{m=2}^{n} \left| \frac{\alpha \delta}{e^{j\omega d} + e^{j\omega(d-1)} + \alpha \delta} \right|^{2(n-m)}$$
$$\cdot \left| \frac{1 - e^{-j\omega}}{e^{j\omega d} + e^{j\omega(d-1)} + \alpha \delta} \right|^2 \sigma_m^2 d\omega$$

The simulation settings are taken as follows: $\mathbb{F} = 3000$, $N = 15$, $\tau = 1$, $\delta = [1\ 0.1\ 0.01]$, $d = \tau/\delta$, $\alpha = 0.5$, $\sigma = 2$, $A_0 = 1$, $\omega_0 = 0.5$. The result shows in Figure A1.

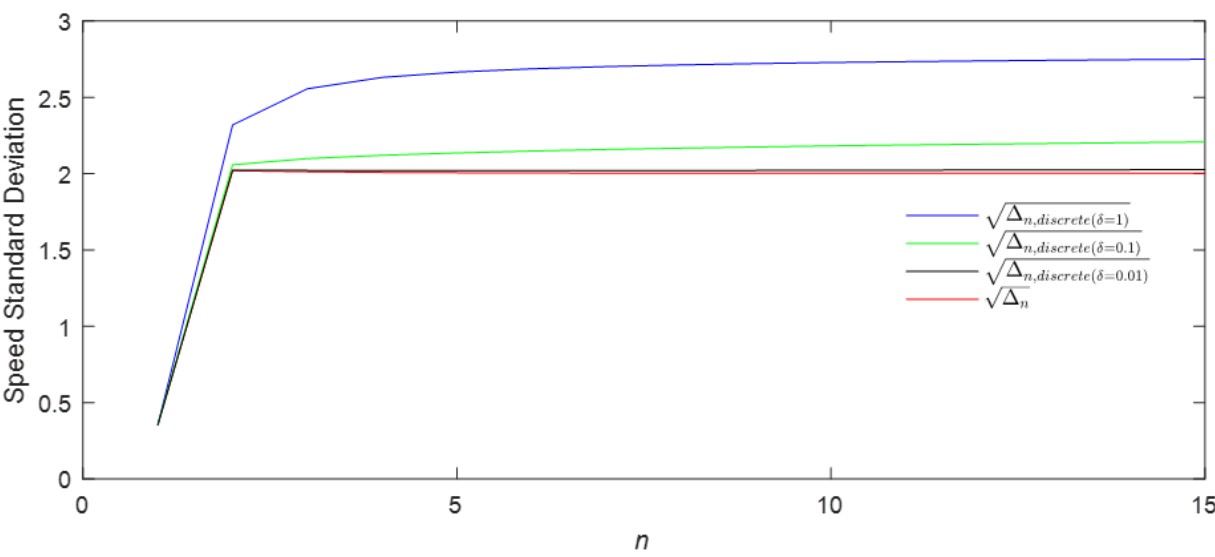

**Figure A1.** Simulation results for discrete and continuous model.

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
