# Peer review of "Stability Analysis of Continuous Stochastic Linear Model"

_sustainability, doi:10.3390/su14053036_

Round 1
Reviewer 1 Report
One can introduce different types of stability for the class of the mathematical models investigated. It is necessary to introduce a definition of stability precisely in the sense that the authors use. It is the first key remark.
Appendix A contains trivial proof of an obvious statement. It must be omit without any damage to understand the article
The truncation of the high frequencies of the Fourier spectrum of formula (3) leads to the appearance of the Gibbs phenomenon. The authors in the work do not discuss how significant this phenomenon in the conditions of the problem investigated.
Applying the Laplace transform to equation (4) the distance p in the right hand part was replace by the Laplace transform from speed V in eq. (6). Why? It is a key remark.
On line 65, enter spacing s is not used further. In formula (5) and beyond, s is a parameter in the Laplace transform (apt remark)
Reviewer 2 Report
This manuscript is very theoretical and should be correlated with field data to match well with the special issue. More literature is required in this regard.
The novelty is not very clear of this research.
Reviewer 3 Report
This paper proposes an extension to the continuous case of a recent stability analysis method in the frequency domain for the growth pattern of traffic oscillations. Overcoming the limitations of the previous study related to the discrete case, the paper proposes the analytical expression for the evolution of speed standard deviation, comparing results using simulation.
I would thank the authors for their work. Below I propose my suggestions, hoping that they can find them useful to improve the paper whose content is certainly interesting in a field of primary interest for transportation and traffic engineering.
- The abstract appears somewhat scarce. In my opinion, the authors must better highlight the research context of the work and explain the value of the approach followed, the methodologies they have used, the results and their potential uses. It is advisable to review the text, emphasizing the contents of the paper and the results obtained.
- The authors presented a brief review of the literature in the introduction. In my opinion, the paper must have a deepened literature framework enriched with discussion contents and related references, which allow the reader to better frame the topic.
- The motivation of the work would seem to be, trivially, only a lack of a previous study, i.e. [19], which is cited and taken as the principal reference for the development of the study. In my opinion, it is advisable to propose a thorough discussion on the research question. The authors can better clarify the research need and the paper purpose, in addition to the gap mentioned and the necessity to fill it, also considering possible applications of the results. Ultimately, these clarifications will allow the paper to enhance the authors' contribution to the research theme through their work.
- Given the number of variables and symbols assumed in the paper, I suggest inserting a table/list of notations. I also recommend a review of variables/symbols in the text and the equations ensuring their homogeneity and clarity. For example, there are some undeclared variables (e.g. an in eq. (1) and eq. (4)) or non-homogeneous notations (e.g. N and ?).
- An overall revision of the text is recommendable. There are typographical errors in wording, missing spaces or punctuation, equations without relative numerical reference, non-homogeneous notation or formatting. There are also some undeclared acronyms (e.g. DLCF and SLCF) that need explanation in the paper.
- In my opinion, the authors can introduce literature references also in sections 2, 3 and 4 of the paper. These references can help the reader deepen the issues taken into consideration in the analyses.
- It is recommendable to improve the quality of the figures and graphs presented in the paper.
- In section 4, it is advisable to clarify how the authors carried out simulations.
- It is advisable to review the conclusions because, in my opinion, the current form does not sufficiently enhance the authors' contribution to the research field gained with their work. My suggestion is to recall the context and aims of the work and then extend the discussion of the results, concerning their novelty compared to previous studies and their usefulness in the research topic, and then clarify any limitations and needs for further development. Visions and suggestions for applying the proposed method and its results in traffic control technologies and measures are highly encouraged.
Round 2
Reviewer 1 Report
The new version of the paper is better that previous one. The article may be of interest for traffic modelling.
Reviewer 2 Report
The manuscript still needs improvement as per my previous comments.
Author Response
The manuscript still needs improvement as per my previous comments.
Previous comments:
Point 1: This manuscript is very theoretical and should be correlated with field data to match well with the special issue. More literature is required in this regard.
Response 1: Thank you for your suggestion, we have added a field data leading vehicle to match the special issue. More literature is added for readers to better frame the topic.
Point 2: The novelty is not very clear of this research.
Response 2: Thank you for your suggestion, we have rewritten abstract, introduction and conclusion to make the novelty of this paper clearer.

Reviewer 3 Report
First of all, thanks to the authors for their new version of the paper, with which they took into consideration the revision suggestions.
Here are a few suggestions for this second round of review.
- A general revision of the editing is recommended, for some errors that appear to be in the new version of the paper (eg line 72, there is repetition for "the paper"; "high-way" line 268; some captions indicate "Figure ", in others" Fig. "appears; ...)
- I recommend revising the numbering of the references according to the order in which they appear in the text of the paper for the first time. In the current version of the paper, there seem to be some problems in the order of the references, in particular starting from [30].
I suggest checking the reference to the paper [34] of the list that is quoted on line 98, verifying that it has not been entered incorrectly, in place of the paper [30] of the list. - I still suggest an improvement of the figures, ensuring that the labels appearing in the graphs are homogeneous throughout the paper in terms of font and size. I also recommend a revision of Figure 3 to avoid overlapping between the legend labels and the data series lines.
- Finally, if "the previous work" in the conclusions (line 257) refers to the paper [30] of the list of references, it is necessary to quote this paper again also in this position.
- In the conclusions, I recommend even more evidence for the possible uses that may arise from this research in traffic control applications.
Author Response
Response to Reviewer 3 Comments
First of all, thanks to the authors for their new version of the paper, with which they took into consideration the revision suggestions.
Here are a few suggestions for this second round of review.
Point 1: A general revision of the editing is recommended, for some errors that appear to be in the new version of the paper (eg line 72, there is repetition for "the paper"; "high-way" line 268; some captions indicate "Figure ", in others" Fig. "appears; ...)
Response 1: Thank you for your suggestion, we have reviewed the paper and corrected the errors.
Point 2: I recommend revising the numbering of the references according to the order in which they appear in the text of the paper for the first time. In the current version of the paper, there seem to be some problems in the order of the references, in particular starting from [30].
I suggest checking the reference to the paper [34] of the list that is quoted on line 98, verifying that it has not been entered incorrectly, in place of the paper [30] of the list.
Response 2: We have checked and corrected the the order of the references as your request.
Point 3: I still suggest an improvement of the figures, ensuring that the labels appearing in the graphs are homogeneous throughout the paper in terms of font and size. I also recommend a revision of Figure 3 to avoid overlapping between the legend labels and the data series lines
Response 3: According to your suggestion, we remade all the figures and now all the fonts and sizes are consistent.
Point 4: Finally, if "the previous work" in the conclusions (line 257) refers to the paper [30] of the list of references, it is necessary to quote this paper again also in this position.
Response 4: Thank you for your suggestion, we changed the manuscript.
Point 5: In the conclusions, I recommend even more evidence for the possible uses that may arise from this research in traffic control applications.
Response 5: Thank you for your suggestion. We proposed several specific measures which may help dampening the highway traffic oscillations according to the conclusion of this paper.